# From the Triage to the Intermediate Area: A Simple and Fast Model for COVID-19 in the Emergency Department

**DOI:** 10.3390/ijerph19138070

**Published:** 2022-06-30

**Authors:** Erika Poggiali, Enrico Fabrizi, Davide Bastoni, Teresa Iannicelli, Claudia Galluzzo, Chiara Canini, Maria Grazia Cillis, Davide Giulio Ponzi, Andrea Magnacavallo, Andrea Vercelli

**Affiliations:** 1Emergency Department, “Guglielmo da Saliceto” Hospital, 29121 Piacenza, Italy; erikapoggiali2@gmail.com (E.P.); dbastonidoc@gmail.com (D.B.); tiannice65@gmail.com (T.I.); claudia88.galluzzo@gmail.com (C.G.); c.canini@ausl.pc.it (C.C.); m.cillis@ausl.pc.it (M.G.C.); d.ponzi@ausl.pc.it (D.G.P.); a.magnacavallo@ausl.pc.it (A.M.); 2DISES & DSS, Università Cattolica del Sacro Cuore, 29122 Piacenza, Italy; enrico.fabrizi@unicatt.it

**Keywords:** COVID-19, SARS-CoV-2, lung ultrasound, emergency department, triage, score

## Abstract

Introduction: The early identification of patients with SARS-CoV-2 infection is still a real challenge for emergency departments (ED). First, we aimed to develop a score, based on the use of the lung ultrasonography (LUS), in addition to the pre-triage interview, to correctly address patients; second, we aimed to prove the usefulness of a three-path organization (COVID-19, not-COVID-19 and intermediate) compared to a two-path organization (COVID-19, non-COVID-19). Methods: We retrospectively analysed 292 patients admitted to our ED from 10 April to 15 April 2020, with a definite diagnosis of positivity (93 COVID-19 patients) or negativity (179 not-COVID-19 patients) for SARS-COV-2 infection. Using a logistic regression, we found a set of predictors for infection selected from the pre-triage interview items and the LUS findings, which contribute with a different weight to the final score. Then, we compared the organization of two different pathways. Results: The most informative factors for classifying the patient are known nasopharyngeal swab positivity, close contact with a COVID-19 patient, fever associated with respiratory symptoms, respiratory failure, anosmia or dysgeusia, and the ultrasound criteria of diffuse alveolar interstitial syndrome, absence of B-lines and presence of pleural effusion. Their sensitivity, specificity, accuracy, and AUC-ROC are, respectively, 0.83, 0.81, 0.82 and 0.81. The most significant difference between the two pathways is the percentage of not-COVID-19 patients assigned to the COVID-19 area, that is, 10.6% (19/179) in the three-path organization, and 18.9% (34/179) in the two-path organization (*p* = 0.037). Conclusions: Our study suggests the possibility to use a score based on the pre-triage interview and the LUS findings to correctly manage the patients admitted to the ED, and the importance of an intermediate area to limit the spread of SARS-CoV-2 in the ED and, as a consequence, in the hospital.

## 1. Introduction

Since the worldwide spread of the SARS-CoV-2 infection in December 2019 [1], several new SARS-CoV-2 variants have been progressively detected in the general population at different times and with different clinical consequences [2], even in vaccinated patients [3]. The challenges posed by these variants identified across the globe is still a matter of debate among epidemiologists and virologists in terms of their effects on therapeutic and prophylactic interventions [2,4].

Piacenza (Emilia-Romagna, Northern Italy) has been one of the Italian epicentres since the first case of COVID-19, recognized in Italy in Codogno (Lombardy) on 21 February 2020. Codogno is very close to Piacenza, approximately 15 km away in the Po valley, which is the most industrialized area of Northern Italy characterized by dense trade (Figure 1). Our hospital is the main hospital with a hub-and-spoke organization, a Radiology Department, and an Intensive Care Unit (ICU) for all the inhabitants of Piacenza and the surrounding four valleys—Val Nure, Val d’Arda, Val Tidone, Val Trebbia (for a total of 287,172 inhabitants at 2019). Within a few days, the COVID-19 epidemic paralysed our public health system and hospital organization, becoming a challenge for our Emergency Department (ED) with several dramatic consequences in the re-organization of the ED and the entire hospital to avoid the complete collapse of our health system [5,6,7]. From 1 March to 18 April 2020, 2665 COVID-19 patients have been admitted to our emergency room; of these admissions, 1675 (63%) patients were hospitalized, and 70 (3%) patients died in the emergency room (Figure 2).

Italy was the first among the Western countries to be involved in the pandemic, with severe and dramatic consequences, particularly in the Po Valley in the Lombardy and Emilia-Romagna territories [8,9,10,11,12,13,14], where most cases and deaths were registered. As reported by Murgante et al. the fatality rates in Lombardy and Emilia-Romagna were significantly higher, respectively, 1.3% vs. 4.5% at *p* < 0.001, compared to the rest of Italy [8]. Social policies, healthcare strategies, climate, air pollution, and geography were the main factors that influenced the differing spread of the novel coronavirus across the Italian territory [8,15,16].

In a very few weeks COVID-19 had become a worldwide threat to health, and emergency clinicians had to learn as much as possible about this dangerous outbreak, separating fact from speculation [17] and trying to find real solutions in such a complex situation [18,19,20,21].

As firstly reported by Poggiali et al. lung ultrasound (LUS) has been crucial in the early management of COVID-19 associated pneumoniae [22], and these preliminary results have been confirmed by several authors as reported in literature [23,24,25,26,27,28,29,30]. Since the beginning of the COVID-19 pandemic, LUS played a key role in the diagnosis and the management of COVID-19 lung injury, showing a high diagnostic accuracy even in the first stage of the disease [27,31]. LUS is a simple widespread technique, mainly applied not only in critical care, emergency medicine, and trauma surgery, but also in pulmonary and internal medicine. As reported by Soldati et al. LUS can be useful in the following situations: triage (pneumonia/non-pneumonia) of symptomatic patients at home as well as in the prehospital phase, diagnostic suspicion and awareness in the emergency department setting, prognostic stratification and monitoring of patients with pneumonia on the basis of the extension of specific patterns and their evolution toward the consolidation phase in the emergency department setting, treatment of intensive care unit patients with regard to ventilation and weaning, monitoring the effect of therapeutic measures (antiviral or others), and reducing the number of health care professionals exposed during patient stratification (a single clinician would be necessary to perform an objective medical examination and imaging investigation directly at the patient’s bed) [32]. For the first time, during the COVID-19 pandemic, international evidence-based recommendations for point-of-care LUS have been proposed by a multidisciplinary panel of 28 experts from eight countries [27], based on a MEDLINE literature search from 1966 to October 2010 in the main topic of LUS, including pneumothorax, interstitial syndrome, lung consolidation, monitoring, neonatology, pleural effusion. Seventy-three specific recommendations are reported in the document [27] with the main aim to provide a rigorous method available to all clinicians for assessing and monitoring lung diseases or injuries, delaying or even avoiding transportation to the radiology suite, and guiding life-saving therapies in extreme emergency such as COVID-19 pandemic.

In patients with COVID-19 pneumonia, LUS reveals a typical pattern of diffuse interstitial lung syndrome, characterized by multiple or confluent bilateral B-lines with spared areas, thickening of the pleural line with pleural line irregularity and peripheral consolidations. As reported by Dacrema et al. [23], LUS findings are consistent with high resolution chest CT scan (HRCT) in correctly screening 130 patients out of 131 COVID-19 pneumoniae cases (99.2%), showing optimal sensitivity (Figure 3). The authors proposed LUS as a simple screening tool for COVID-19 pneumonia diagnosis in the context of outbreak burst areas where prompt isolation of suspected patients is crucial for patients’ and operators’ safety.

Despite the vaccines’ development and their potential to prevent disease transmission in a larger population, it is noteworthy that the percentage of asymptomatic infected patients is about 30–40% in Piacenza, as reported by the contact tracing of the local public health system. As a consequence, unrecognized positive patients can be addressed to the “COVID-19 free area” and put negative patients at risk of infection. Based on these observations, we needed to develop a strategy to avoid the spread of the SARS-CoV-2 in the “clean areas” of the ED and as a reflection, of the hospital. During the “phase 2” of the COVID-19 Italian epidemic, we designed a pre-triage decision making-process, based on the pre-triage interview combined with LUS performed in the triage area, confirming the pivotal role of the triage to promptly identify COVID-19 patients even in absence of typical signs and symptoms of the infection [33]. Our study failed to demonstrate the utility of LUS because of the small size of the patient sample that we investigated in the study; we designed a new safety protocol to identify COVID-19 patients promptly after their arrival in the triage, and we created different pathways and areas in the ED to curb the risk of transmission as much as possible and, as a consequence, to avoid the development of outbreaks in the hospital. 

Herein, we discuss how we developed a new model to correctly manage patients in the ED based on a score calculated in the triage area, and three different pathways: COVID-19, not COVID-19 and intermediate.

## 2. Materials and Methods

This is a retrospective observational study including 451 patients aged over 13-year-old, consecutively admitted to the ED of “Guglielmo da Saliceto” Hospital in Piacenza, Emilia-Romagna, Italy, from 10 April to 15 April 2020. The study was approved by the local ethics committee (“Comitato Etico AVEN dell’Area Vasta Emilia Nord”, Protocol Number 2020/60883, Registry Number 601/2020/OSS. No animal was used for research proposal).

### 2.1. Organization of the Emergency Room: Pre-Triage and Triage Area

Before being admitted to the ED, all the patients were interviewed by an expert nurse in the pre-triage area, located outside the hospital, close to the entrance of the ED, using a standard schedule to investigate clinical and epidemiological criteria for COVID-19. If the patient was evaluated by the local pre-hospital Emergency Medicine System, the medical or nursing team had to fill in the same schedule adopted in the pre-triage area for each patient before their arrival to the ED. 

The clinical criteria were:isolated fever (body temperature > 37 °C)fever and respiratory symptomsfever and gastrointestinal symptomsvomiting or diarrheageneral malaisemyalgiasanosmia and dysgeusiarespiratory failure, defined as peripheral oxygen saturation (SpO2) less than 95% at room ambient, or 92% for patients with chronic obstructive pulmonary diseases (COPD).

The epidemiological criteria were: a close contact in the last 14 days with a subject with a positive nasopharyngeal (NP) swab for SARS-CoV-2, defined as follows: for at least 15 min at a distance less than 2 m without proper personal protective equipmenthome isolation for suspected SARS-CoV-2 infectiona NP swab positive for SARS-CoV-2reported death of a close family member for unknown causesall the patients living in a retirement home.

At the time of the enrollment of the patients, a case could be considered as “negativized” if asymptomatic and with two negative RT-PCR NP swabs at a distance of 24 h from each other, at least 14 days after the positive swab.

In the triage area, all the patients were investigated using a rapid and simple LUS protocol to explore all the pulmonary fields from the apex to the bases, both anterior and posterior, as follows: 2 anterior, 2 posterior and 2 lateral (total 12 areas). LUS was performed using an ultrasound machine as follows: ESAOTE MyLab Alpha; ESAOTE Mylab 30 Gold equipped with a convex probe, probe AC2541 (frequency range 1–8 MHz), probe CA631 (frequency range 1–8 MHz), respectively, (Esaote Medical Systems, Florence, Italy); Philips Afniti 70 equipped with convex probe C6-2 (range frequency 2–6 MHz) (Philips N.V., Amsterdam, The Netherlands); Samsung HM70a equipped with convex probe CA1-7AD (range frequency 1–7 MHz) (Samsung Medison, Republic of Korea). All the LUS were performed by a member of the medical team of our ED, including those professionals who were considered inexperienced in LUS, i.e., with less than 6 months of experience. Based on the observation that each pulmonary area should be scanned for 10 s, the mean time to perform a complete and correct LUS was 2 min. 

LUS findings could be:diffuse alveolar-interstitial syndrome (AIS) defined as three or more separate or coalescent B-lines in at least two fields for each hemithorax [34]focal AIS defined as three or more B-lines in a fieldnormal pulmonary pattern (A pattern)pleural effusion.

### 2.2. Classification of the Patients and Data Collection

All the data (clinical and epidemiological criteria, LUS, radiological and laboratory findings including RT-PCR NP swab) were collected anonymously in a database. 

An RT-PCR NP swab was performed in all the patients to be admitted, or in those with a strong clinical suspicious of SARS-CoV-2 infection. In our experience, a positive NP swab is sufficient to define the patient as “positive”, but a negative NP swab in presence of highly suspected clinical and radiological criteria cannot exclude COVID-19 [35]. To clearly define a patient as “negative”, a negative RT-PCR NP (NP) swab is mandatory but not sufficient, due to the high frequency of false negative and asymptomatic carriers. 

As consequence, we defined as “positive” for COVID-19, a patient with:a RT-PCR NP swab positive for SARS-CoV-2 in the past 14 days, at admission in the ED or within 72 h, ora clinical criterion and signs of interstitial pneumonia at chest X-ray or high-resolution CT-scan even with negative RT-PCR NP swab.

We defined as “negative”, a patient with a negative RT-PCR NP swab at admission in the ED or within 72 h, and:neither clinical nor epidemiological criteria nor radiological signs of COVID-19 pneumonia, orthe absence of antibodies (IgG) against SARS-CoV-2 in a period between 1 and 3 months after discharge oran alternative diagnosis to SARS-CoV-2 infection as justification for symptoms, after hospitalization of at least one week.

We defined as “indeterminate”, a patient who did not meet neither the positive nor the negative criteria.

### 2.3. Statistical Analysis

Patients with an undetermined final diagnosis or not investigated with LUS were excluded from statistical analysis.

Among the many possible alternatives, we considered logistic regression as a classifier, since its additive structure reproduces the logic of scoring construction adopted by the ward’s staff. We selected a smaller set of predictors from the initial one using a backward stepwise algorithm (based on AIC as model comparison tool). 

The logistic regression algorithm is used to express a response variable as a function of a set of possible predictors and idiosyncratic residuals when the response variable is binary. It is similar to linear regression where the mean response variable, say E(y)*,* is expressed a linear function of possible predictors $x_{1},\ldots ,x_{p}$, i.e., $E\left(y\right)=\beta_{0}+x_{1}\beta_{1}+\ldots +{\;x}_{p}\beta_{p}$. Nonetheless, it is different as when y is binary (0/1) its expectation is a probability (E(y)=P(y=1)=π and as such a number in the [0,1] interval. A linear combination such as $\beta_{0}+x_{1}\beta_{1}+\ldots +{\;x}_{p}\beta_{p}\;$ could generate predicted values outside this range. For this reason a transformation of E(y) is used instead, namely, we have $logit\left(E\left(y\right)\right)=\beta_{0}+x_{1}\beta_{1}+\ldots +{\;x}_{p}\beta_{p}$ where logit(.) stays for the logarithm of the probability odd of E(y) i.e., logit{E(y)}=log{E(y)/[1−E(y)}. The $\beta_{0},\ldots ,\;\beta_{p}$ parameters are unknown and estimated from the data using an algorithm known as “iterative least square” to maximize the data likelihood. Each $\beta_{j}$ describes changes when $x_{j}$ increases by 1 unit, all the remaining predictors are kept constant; $\exp\left(\beta_{j}\right)\;$ describes how the odd E(y)/{1−E(y)}  changes under the same conditions [36].

We assessed the predictive power of the model by using the overall accuracy, sensitivity and specificity rates, along with the area under the receiver-operator characteristic curve (AUC-ROC) and Tjur’s R square [37]. To avoid any optimistic bias, all statistics were not computed in a sample but using a 10-fold cross-validation algorithm [38]. As for classification, we selected a threshold aimed at striking the best balance between specificity and sensitivity. Specifically, we adopted the Youden’s J statistic [39].

Moreover, we proposed a classification in three groups, aimed at isolating a so called “grey” or “intermediate area”. This classification is based on the same criterion, under the constraint of limiting the intermediate group up to 30% of the observed cases.

Since variables related to LUS are of special relevance in our analysis, we used a Fisher exact test to assess the significance of the odds-ratio associated to the presence of diffuse AIS without pleural effusion as predictor of COVID-19 diagnosis in patients with respiratory failure. We repeated the same analysis also for the presence of A pattern with or without pleural effusion and pleural effusion without diffuse AIS as predictors of no COVID-19 diagnosis in patients with respiratory failure.

All computations are based on the software R, version 4.0.2 (R Core Team (2020). R: A language and environment for statistical computing. R Foundation for Statistical Computing, Vienna, Austria. URL: https://www.R-project.org/ (accessed on 1 June 2020)) and the packages brglm2 [40] and performance [41].

## 3. Results

Four hundred and fifty-one patients were enrolled in the study, of whom SARS-CoV-2 infection status could be established in 295 (65%) cases. Of these, LUS was performed in triage in 272 (60%) patients constituting the sample analysed. The COVID-19 patients were numbered at 93, and 179 resulted as being not-COVID-19 patients. Among the COVID-19 patients, 69 (74%) were identified by the positive NP swab, and 24 (25%) by clinical and radiological criteria (Figure 3). Therefore, the prevalence of disease in the sample was 34%. The clinical characteristics of the patients of the sample are shown in Table 1.

The LUS findings of the sampled patients are shown in Table 2.

Considering the subgroup of patients with respiratory failure, 35 (57%) were COVID-19 patients and 26 (46%) were not-COVID-19 patients, as reported in Table 1. The results of the LUS in this subgroup of patients are as shown in Table 3.

From the logistic regression, the epidemiological, clinical and ultrasound criteria significant at the 5% level are: a previous positive NP swab, precautionary isolation status, coming from a nursing home, fever associated with respiratory symptoms; diffuse AIS is weakly significant. Tjur’s R2 for the model is 0.50, with sensitivity 0.82, specificity 0.85, accuracy 0.84 (misclassification rate = 0.16); AUC-ROC is 0.82. The presence of numerous statistically not-significant criteria and a rather serious level of collinearity among the predictors, especially among the ultrasound predictors, recommended a regressor selection operation: the presence of collinearity among the regressors means regressors that would actually be significant, do not appear. We, therefore, arrived at a model that includes fewer predictors: known NP swab positivity, having had close contact, fever associated with respiratory symptoms, respiratory failure, anosmia/dysgeusia, and the ultrasound criteria of diffuse AIS, absence of B-lines, and the presence of pleural effusion are the most informative factors for classifying the patient. The results are shown in Table 4. Tjur’s R2 for the model is 0.47, with sensitivity 0.83, specificity 0.81, accuracy 0.82; the AUC-ROC is 0.81.

The Table 5 shows the scores as attributed by the logistic regression algorithm. In addition to the criteria selected by the logistic regression, isolated fever and fever associated with gastrointestinal symptoms were also included in the score (Figure 4).

Applying the score in a system that foresees two paths, COVID-19 and not-COVID-19, on the basis of the optimal threshold (chosen by optimising the Youden index, i.e., the sum of sensitivity and specificity), patients with a score ≥ 0.6 are to be directed to the COVID-19 area and patients with a score < 0.7 to the not-COVID-19 area. The distribution of patients would be as shown in Table 6.

In a classification of patients into three areas, trying to minimize both COVID-19 patients in the not-COVID-19 area and not-COVID-19 patients in the COVID1-9 area, and assuming that the intermediate area is not assigned more than 30% of the patients, patients with score 0 or less are assigned to the not-COVID-19 area, those with score > 0.8 to the COVID-19 area, and those with score 0.1–0.8 to the intermediate area. The allocation of patients to the three areas is shown in Table 7.

Comparing the results of assignment to 2 or 3 pathways, the percentage of not-COVID-19 patients assigned to the COVID-19 area increased from 18.9% (34/179) to 10.6% (19/179) respectively, and it was statistically significant (*p* = 0.037). 

The risk of a not-COVID-19 patient having contact with a COVID-19 patient in the not-COVID-19 area decreased from 9.9% (16/161) to 7.4% (8/107). In this case, the risk reduction is not statistically significant. The risk for a not-COVID-19 patient in the intermediate area to come into contact with a COVID-19 patient is 23% (19/80). 

Ultrasound performance improves when considering the subgroup of patients with respiratory failure. The patients with respiratory failure undergoing LUS are 61, of whom 35 are COVID-19 patients and 26 are not-COVID-19 patients. Assuming diffuse AIS without pleural effusion (positive LUS) is a predictor of COVID-19, the results are shown in Table 8.

The presence of diffuse AIS without pleural effusion in a patient with respiratory failure has a sensitivity and specificity for the diagnosis of COVID-19 of 74% and 77%, respectively, with an odds ratio of 9.2 (95% CI 2.6–36.9) (*p* < 0.00001). 

Conversely, assuming that the presence of an A pattern or pleural effusion with A pattern or B-lines that do not represent diffuse AIS (positive ultrasound) are predictors of the absence of COVID-19, the results are reported in Table 9.

The presence of an A pattern or a pleural effusion without diffuse AIS in a patient with respiratory failure has a sensitivity and specificity for the diagnosis of absence of COVID-19 of 38% and 97%, respectively, with an odds ratio of 0.04 (C.I. at 95% 0.001–0.376) (*p* = 0.00003).

## 4. Discussion

LUS is a harmless test with an immediate response, which can be performed in a few minutes with any type of ultrasound machine at the patient’s bedside, even by inexperienced operators, as it is easy to learn [24]. In this study, clinicians with less than six months’ experience in ultrasound were chosen, both because of the scarcity of human resources during the COVID-19 pandemic and to overcome the limitation of the operator dependency, which is often attributed to ultrasound. 

Statistical analysis showed that clinical and epidemiological criteria alone would not be able to classify patients correctly. In particular, LUS proved useful in better identifying a group of patients with a very low probability of being COVID-19 patients: the most significant ultrasound findings for this purpose were the presence of pleural effusion and the absence of B-lines. Both criteria contribute to the score with a negative value in patients with negative epidemiological criteria and without particularly suspicious clinical symptoms, such as fever with respiratory symptoms or anosmia or dysgeusia, which allows them to be placed in the not-COVID-19 area. Thus, considering the whole sample, LUS tends to increase the specificity of the method rather than the sensitivity.

The performance of LUS is further improved when considering the subgroup of patients with respiratory failure. In COVID-19 patients with respiratory failure is almost always caused by interstitial pneumonia (pulmonary embolism is another cause, but is usually associated with interstitial pneumonia), for which LUS has a high sensitivity. As reported in studies in which LUS was applied in the ED for the diagnosis of SARS-COV-2 pneumonia, sensitivity values ranged from 52% to 92%, and specificity values from 65% to 89%, depending on the gold standard used for diagnosis and LUS score threshold used to consider the test positive: the typical finding is diffuse B-lines in several lung fields, sometimes confluent as observed in “white lung” [29,42,43,44].

In our case series, the presence of diffuse AIS without pleural effusion in a patient with respiratory failure has a strong association with the diagnosis of COVID-19 (odds ratio 9.2). Even stronger (odds ratio 0.04) is the association between the absence of disease and the presence of one of the following ultrasound patterns: A pattern without pleural effusion, A pattern with pleural effusion, pleural effusion with B-lines to an extent that does not constitute diffuse AIS.

During the first COVID-19 wave, there were no rapid diagnostic tests that could be used in triage as screening tests for SARS-CoV-2 infection. The capacity of the laboratory to perform a limited number of molecular NP swabs and the reporting time (in urgent cases the result was available in 4–6 h) prevented its widespread use by all or most patients admitted to the ED, and certainly made it unsuitable for screening patients in triage.

Currently, antigenic tests have reached a level of accuracy to replace the molecular swab in the majority of cases. Even the most accurate tests (third-generation antigenic tests) require only a few minutes to provide a result, so they are well suited for use in triage to direct patients more correctly to COVID-19 and not-COVID-19 areas in the ED. However, the large-scale use of these tests results in very high costs, which may prevent their application in certain settings. Furthermore, although the sensitivity of these tests is over 90%, the risk of false negatives and intra-hospital infections does not disappear. For these reasons, we are convinced that LUS can still play a role in the early detection of COVID-19 patients. Even with the addition of LUS to the triage interview, a significant proportion of patients who are misclassified still remains: out of 111 patients who are referred to the COVID-19 area, 34 (18.9%) are not-COVID-19; and the risk for a not-COVID-19 patient to come into contact with a COVID-19 patient in the not-COVID-19 area is 9.9%, resulting in an increased risk of personal contagion for the former and a high risk of generating intra-hospital outbreaks from the not-COVID-19 area. In order to try to contain these risks, we proposed to identify a third so-called intermediate route. By definition, the intermediate area is a hybrid area where COVID-19 and not-COVID-19 patients are present. As a consequence, both clinicians, nurses and patients must be provided with appropriate personal protective equipment, and droplet isolation between patients is mandatory. In order to be able to offer these guarantees, the number of patients entrusted to this area must be limited and appropriate to the facilities available: thinking about the space available in our ED, we established that the number of entrustments in this area should not exceed 30% of the total, and we simulated the distribution of patients by applying the proposed score. This significantly decreased the risk of referring a not-COVID-19 patient to the COVID-19 area (from 18.9 to 10.6%); however, the reduction in the risk of a not-COVID-19 patient coming into contact with a COVID-19 patient in the not-COVID-19 area was statistically insignificant. The stay of a not-COVID-19 patient in a COVID-19 area undoubtedly increases the risk of infection for the patient, which means that even if the patient results are negative for the virus, the patient will have to observe a quarantine period for the exposure. Containing the number of not-COVID-19 patients entering the COVID-19 area offers a clear advantage to the patients and to the clinicians for their subsequent management.

## 5. Conclusions

Based on our experience, LUS is a useful and easy test to enhance the ability of the triage to correctly refer patients to a COVID-19 or not-COVID-19 area. Despite the current availability of rapid antigenic tests as screening tests, and given the expected high cost of large-scale use of these tests, LUS should be considered as a valuable tool to improve triage appropriateness, particularly under conditions of scarce resources in terms of both personnel and space in the EDs. Based on our results, LUS findings could be used together with the pre-triage interview as a useful score to classify patients at admission in the ED. Our study demonstrates the importance of the creation of three flows to correctly manage the patients, as follows: COVID-19, not COVID-19 and intermediate. We are aware that the existence of an intermediate area in the ED requires an additional space and a dedicated staff, but it allows the risk of intra-hospital contagion to be limited as much as possible, especially if effective screening tests, such as antigenic tests, cannot be used in triage. 

## Figures and Tables

**Figure 1 ijerph-19-08070-f001:**
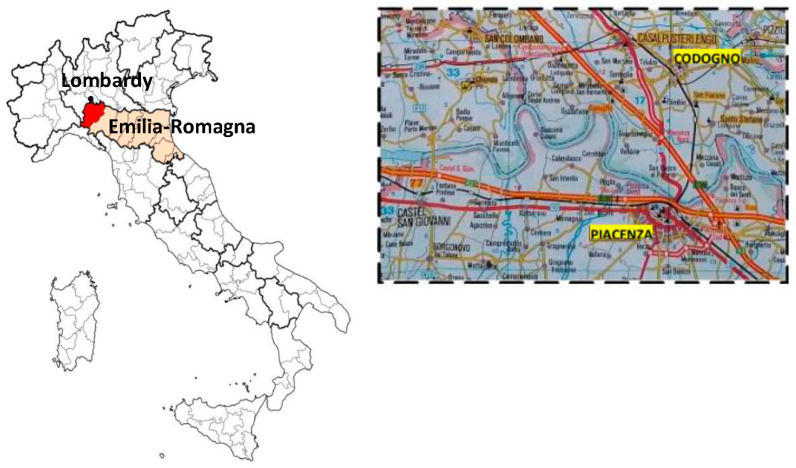
Piacenza and the surrounding four valleys-Val Nure, Val d’Arda, Val Tidone, Val Trebbia (red zone) (Emilia-Romagna, Northern Italy). Codogno (Lombardy) is the black point, 15 km away from Piacenza.

**Figure 2 ijerph-19-08070-f002:**
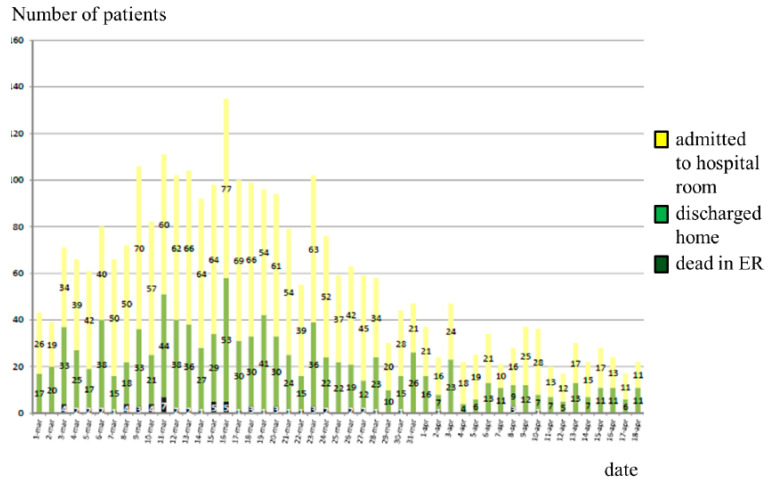
Patients admitted to the ED of Piacenza during the first wave of COVID-19 Italian epidemic (data from 1 March 2020 to 18 April 2020).

**Figure 3 ijerph-19-08070-f003:**
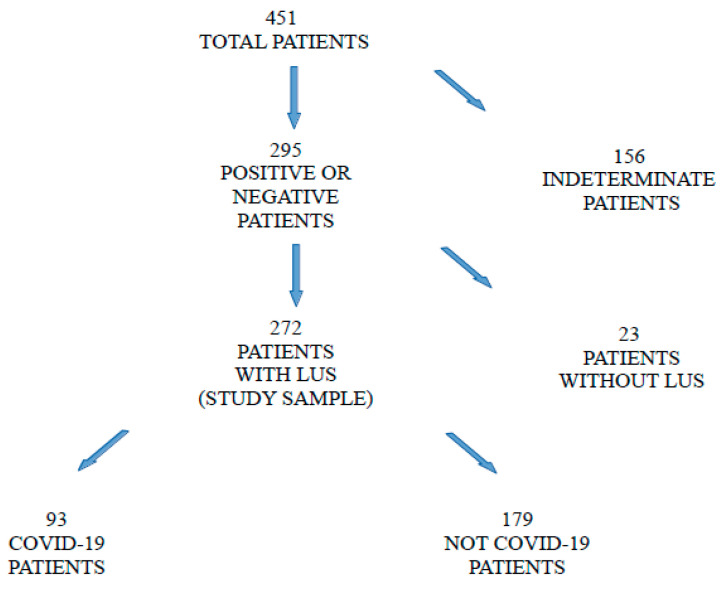
Flow-chart of the study.

**Figure 4 ijerph-19-08070-f004:**
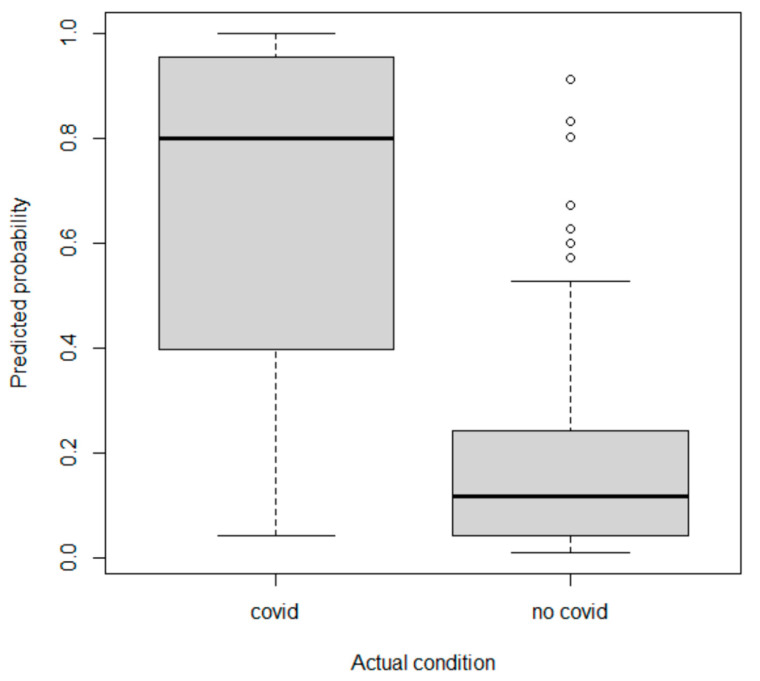
Probability of being a COVID-19 patient predicted by the logistic regression model is plotted against the actual patient status.

**Table 1 ijerph-19-08070-t001:** Patients’ demographic and clinical characteristics, and radiological tests performed in the ED on admission. M, male. F, female. HRCT, high-resolution computed tomography. Respiratory failure is defined as SpO2 less than 95% at room ambient, 92% for patients with COPD.

	COVID-19 Patients (n = 93)	Not-COVID-19 Patients (n = 179)
Sex (M/F)	39 (42%)/54 (58%)	89 (50%)/90 (50%)
Mean age (years) (min-max)	70 (27–96)	61 (15–99)
Fever alone	24 (26%)	21 (12%)
Fever and respiratory symptoms	34 (36%)	20 (11%)
Fever and gastrointestinal symptoms	19 (20%)	20 (11%)
Vomiting or diarrhea	14 (15%)	35 (20%)
General malaise	25 (27%)	27 (15%)
Myalgias	18 (19%)	18 (10%)
Anosmia or dysgeusia	11 (12%)	12 (7%)
Acute respiratory failure	35 (38%)	26 (14%)
Chest X-ray	20 (21%)	61 (34%)
HRCT	67 (72%)	81 (45%)

**Table 2 ijerph-19-08070-t002:** LUS findings in COVID-19 and not-COVID-19 patients. AIS, alveolar-interstitial syndrome. B-lines means focal AIS or B-lines without criteria for diffuse AIS.

	COVID-19 Patients (n = 93)	Not-COVID-19 Patients (n = 179)
A pattern	16 (17%)	91 (51%)
B-lines	31 (33%)	56 (31%)
Diffuse AIS	45 (48%)	24 (13%)
Pleural effusion	9 (9%)	21 (12%)

**Table 3 ijerph-19-08070-t003:** LUS findings in the subgroup of patients with respiratory failure. AIS, alveolar-interstitial syndrome. B-lines means focal AIS or B-lines without criteria for diffuse AIS.

	COVID-19 Patients (n = 35)	Not-COVID-19 Patients (n = 26)
A pattern	1 (3%)	5 (19%)
B-lines	6 (17%)	12 (46%)
Diffuse AIS	28 (80%)	7 (27%)
Pleural effusion	3 (9%)	7 (27%)

**Table 4 ijerph-19-08070-t004:** Results from the logistic regression analysis: coefficients on the log-odd scale and associated standard errors, significance tests. NP, nasopharyngeal. AIS, alveolar-interstitial syndrome. Significance codes: 0 ‘***’ 0.001 ‘**’ 0.01 ‘*’ 0.05 ‘.’ 0.1 ‘ ’ 1.

	Estimate Std.	Error	z Value	Pr(>|z|)
(Intercept)	−1.5128	0.3280	−4.612	0.00000399 ***
NP swab	5.3990	1.5824	3.412	0.000645 ***
Close contact	2.4112	0.5455	4.420	0.00000986 ***
Fever alone	0.4959	0.4548	1.090	0.275509
Fever and gastrointestinal symptoms	−0.1680	0.5268	−0.319	0.749807
Fever and respiratory symptoms	1.2985	0.4620	2.811	0.004943 **
Respiratory failure	0.6819	0.4163	1.638	0.101401
Anosmia and dysgeusia	0.9937	0.5752	1.728	0.084056 .
Diffuse AIS	0.841	0.4151	2.027	0.042671 *
A pattern	−1.6403	0.4911	−3.340	0.000838 ***
Pleural effusion	−0.7286	0.5538	−1.316	0.188257

**Table 5 ijerph-19-08070-t005:** Scores based on the logistic regression classifier. NP, nasopharyngeal. AIS, alveolar-interstitial syndrome.

	Score
Known positive NP swab	5.4
Close contact	2.4
Fever alone	0.5
Fever and gastrointestinal symptoms	0.1
Fever and respiratory symptoms	1.3
Respiratory failure	0.7
Anosmia and dysgeusia	1.0
Diffuse AIS	0.8
A pattern	−1.6
Pleural effusion	−0.7

**Table 6 ijerph-19-08070-t006:** Confusion matrix based on the logistic regression (binary classification). By row the actual patient status, by column their classification according to the algorithm.

	Not COVID-19 Area	COVID-19 Area
Not-COVID-19 patients	145	34
COVID-19 patients	16	77

**Table 7 ijerph-19-08070-t007:** Confusion matrix based on the logistic regression (classification in three groups). By row the actual patient status, by column their classification according to the algorithm.

	Not COVID-19 Area	Intermediate Area	COVID-19 Area
Not-COVID-19 patients	99	61	19
COVID-19 patients	8	19	66

**Table 8 ijerph-19-08070-t008:** 2 × 2 contingency table for LUS, considered positive in the presence of diffuse AIS without pleural effusion.

	Respiratory Failure		
	COVID-19	Not COVID-19	Total
Positive LUS	26	6	32
Negative LUS	9	20	29
Total	35	26	

**Table 9 ijerph-19-08070-t009:** 2 × 2 contingency table for LUS, considered positive in presence of A pattern or pleural effusion with A pattern or B-lines that do not represent diffuse AIS.

	Respiratory Failure		
	COVID-19	Not COVID-19	Total
Positive LUS	10	1	11
Negative LUS	15	34	49
Total	26	35	

## Data Availability

All data are available upon reasonable request to a.vercelli@ausl.pc.it.

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
