# Peer review of "From the Triage to the Intermediate Area: A Simple and Fast Model for COVID-19 in the Emergency Department"

_ijerph, 2022, doi:10.3390/ijerph19138070_

Round 1

Reviewer 1 Report

The authors have investigated the usefulness of lung ultrasound in combination with standard diagnostic methods in order to assess its potential as indicator of early COVID-19 ΄infection diagnosis. The results presented support the use of such combination, however, 1) Potentially there needs to be further assessment of their data using archival n/p samplesauthors have investigated the usefulness of lung ultrasound in combination with standard diagnostic methods in order to assess its potential as indicator of early COVID-19 ΄infection diagnosis. The results presented support the use of such combination, however, potentially there needs to be further assessment of their data using archival more n/p samples

Author Response

The authors have investigated the usefulness of lung ultrasound in combination with standard diagnostic methods in order to assess its potential as indicator of early COVID-19 ΄infection diagnosis. The results presented support the use of such combination, however, 1) Potentially there needs to be further assessment of their data using archival n/p samples authors have investigated the usefulness of lung ultrasound in combination with standard diagnostic methods in order to assess its potential as indicator of early COVID-19 ΄infection diagnosis. The results presented support the use of such combination, however, potentially there needs to be further assessment of their data using archival more n/p samples.

Reply: Thanks for the comments. The role of LUS in the early diagnosis of COVID-19 pneumoniae has been firstly reported in our letter published in Radiology (Poggiali, Erika et al. “Can Lung US Help Critical Care Clinicians in the Early Diagnosis of Novel Coronavirus (COVID-19) Pneumonia?.” Radiology vol. 295,3 (2020): E6. doi:10.1148/radiol.2020200847) and confirmed by several articles (262 citations, www.semanticscholar.org). We have added in the section “introduction” the following references to confirm our claims:

  • Caroselli, Costantino et al. “Early Lung Ultrasound Findings in Patients With COVID-19 Pneumonia: A Retrospective Multicenter Study of 479 Patients.” Journal of ultrasound in medicine : official journal of the American Institute of Ultrasound in Medicine, 10.1002/jum.15944. 18 Jan. 2022, doi:10.1002/jum.15944
  • Abhilash, Kundavaram P P et al. “Acute management of COVID-19 in the emergency department: An evidence-based review.” Journal of family medicine and primary care 11,2 (2022): 424-433. doi:10.4103/jfmpc.jfmpc_1309_21
  • Jackson, Karl et al. “Lung ultrasound in the COVID-19 pandemic.” Postgraduate medical journal 97,1143 (2021): 34-39. doi:10.1136/postgradmedj-2020-138137
  • Zhang, Yao et al. “Lung Ultrasound Findings in Patients With Coronavirus Disease (COVID-19).”  American journal of roentgenologyvol. 216,1 (2021): 80-84. doi:10.2214/AJR.20.23513
  • Sorlini, Cristina et al. “The role of lung ultrasound as a frontline diagnostic tool in the era of COVID-19 outbreak.” Internal and emergency medicine 16,3 (2021): 749-756. doi:10.1007/s11739-020-02524-8

Reviewer 2 Report

This paper could provide interesting inputs to tackle covid-19 and other health emergencies but currently it seems to have several weak points that should be considered and reduced.

Above all, the introduction is very synthetic and quickly says that "Piacenza (Emilia-Romagna, Northern Italy) has been one of the Italian epicentres since the beginning of the pandemic at the end of February 2019 with several and dramatic consequences in the reorganization of the emergency department". Please provide some specific data which can quantify this affirmation also in a comparative perspective with other Italian provinces. Moreover a map able to geolocalise Piacenza in the context of North-East Italy would be useful for foreign readers. Furtherly many works should be cited and contextualized about Italian regions differences and possible reasons. See for example:

https://www.mdpi.com/2071-1050/12/12/5064

http://www.j-reading.org/index.php/geography/article/view/258

https://www.sciencedirect.com/science/article/pii/S0013935120313566

https://rosa.uniroma1.it/rosa03/semestrale_di_geografia/article/view/17031

https://www.elsevier.com/books/mapping-the-epidemic/casti/978-0-323-91061-3

The problematic situations of ED, in Italy and other countries, should be better contextualized, both in ordinary and extraordinary conditions. Please see at least the following contributions for a critical analysis and discussion.

https://www.ncbi.nlm.nih.gov/pmc/articles/PMC2831731/

https://www.mdpi.com/2220-9964/9/10/579

https://www.ncbi.nlm.nih.gov/pmc/articles/PMC5377968/

Then, I think that it would be useful in the introduction, as first discussion, to specify that the method here advanced can be useful in the first phases of similar health emergencies, because at the current state the use of swab is so widespread that generally people arrive at ED already knowing if they are positive at covid-19.

I see that only one figure is used in the paper. I suggest to add some other figures which can facilitate the reading and summarise the data, at the moment reported only in Tables.

The paragraph about results should be improved with some fluent considerations useful to better explain the aims and results reached and the possible benefits for the future researches, starting from the data here presented. Moreover, generally the data are reported in Tables often left to the readers' interpretation, saying "results are shown in Table".

I think that these aspects can be useful to increase the quality of presentation and scientific soundness.

Finally, in terms of references, some lacks seem to be present. I have suggested some exemplifying works which could be useful to broaden the views and provide further rigour to the treatise.

Author Response

Comments and Suggestions for Authors

This paper could provide interesting inputs to tackle covid-19 and other health emergencies but currently it seems to have several weak points that should be considered and reduced.

Above all, the introduction is very synthetic and quickly says that "Piacenza (Emilia-Romagna, Northern Italy) has been one of the Italian epicentres since the beginning of the pandemic at the end of February 2019 with several and dramatic consequences in the reorganization of the emergency department". Please provide some specific data which can quantify this affirmation also in a comparative perspective with other Italian provinces. Moreover a map able to geolocalise Piacenza in the context of North-East Italy would be useful for foreign readers.

Reply: We thank the reviewer for this comment. We have added all the required data in the section “Introduction” (see the text tacked in red, and the figure 1 and 2).

Comments and Suggestions for Authors

This paper could provide interesting inputs to tackle covid-19 and other health emergencies but currently it seems to have several weak points that should be considered and reduced.

Above all, the introduction is very synthetic and quickly says that "Piacenza (Emilia-Romagna, Northern Italy) has been one of the Italian epicentres since the beginning of the pandemic at the end of February 2019 with several and dramatic consequences in the reorganization of the emergency department". Please provide some specific data which can quantify this affirmation also in a comparative perspective with other Italian provinces. Moreover a map able to geolocalise Piacenza in the context of North-East Italy would be useful for foreign readers.

Reply: We thank the reviewer for this comment. We have added all the required data in the section “Introduction” (see the text tacked in red, and the figure 1 and 2).

Figure 1. Piacenza and the surrounding four valleys - Val Nure, Val d’Arda, Val Tidone, Val Trebbia (red zone) (Emilia-Romagna, Northern Italy).     

Figure 2. Patients admitted to the ED of Piacenza during the first wave of COVID-19 Italian epidemic (data from 1st March 2020 to 18th April 2020).

Furtherly many works should be cited and contextualized about Italian regions differences and possible reasons. See for example:

https://www.mdpi.com/2071-1050/12/12/5064

http://www.j-reading.org/index.php/geography/article/view/258

https://www.sciencedirect.com/science/article/pii/S0013935120313566

https://rosa.uniroma1.it/rosa03/semestrale_di_geografia/article/view/17031

https://www.elsevier.com/books/mapping-the-epidemic/casti/978-0-323-91061-3

Reply: We have added some of the suggested references in the Introduction.

The problematic situations of ED, in Italy and other countries, should be better contextualized, both in ordinary and extraordinary conditions. Please see at least the following contributions for a critical analysis and discussion.

https://www.ncbi.nlm.nih.gov/pmc/articles/PMC2831731/

https://www.mdpi.com/2220-9964/9/10/579

https://www.ncbi.nlm.nih.gov/pmc/articles/PMC5377968/

Reply: We have added the suggested 3 references in the introduction.

Then, I think that it would be useful in the introduction, as first discussion, to specify that the method here advanced can be useful in the first phases of similar health emergencies, because at the current state the use of swab is so widespread that generally people arrive at ED already knowing if they are positive at covid-19.

Reply: The same observation is present in the discussion, as follows: “During the first COVID-19 wave, there were no rapid diagnostic tests that could be used in triage as screening tests for SARS-CoV-2 infection. The capacity of the laboratory to perform a limited number of molecular NP swabs and the reporting time (in urgent cases the result was available in 4-6 hours) prevented its widespread use by all or most patients admitted to the ED, and certainly made it unsuitable for screening patients in triage. Currently, antigenic tests have reached a level of accuracy to replace the molecular swab in the majority of cases. Even the most accurate tests (third generation antigenic tests) require only a few minutes to provide a result, so they are well suited for use in triage to direct patients more correctly to COVID-19 and not-COVID-19 areas in the ED. However, the large-scale use of these tests results in very high costs, which may prevent their application in certain settings. Furthermore, although the sensitivity of these tests is over 90%, the risk of false negatives and intra-hospital infections does not disappear. For these reasons……..”. For this reason, in order to avoid a repetition in the introduction, we have decided not to modify the introduction.

I see that only one figure is used in the paper. I suggest to add some other figures which can facilitate the reading and summarise the data, at the moment reported only in Tables.

Reply: Considering all the revisions requested by the 3 reviewers, there are 4 figures and 9 tables. The choice of using “table” instead of “figure” in the results is due to the results’ characteristics: patients’ sample (table 1), LUS findings (tables 2 and 3), statistical analysis and results (table 4-9). We do not agree to modify the tables in figures that are not the best way to show this kind of results.

The paragraph about results should be improved with some fluent considerations useful to better explain the aims and results reached and the possible benefits for the future researches, starting from the data here presented. Moreover, generally the data are reported in Tables often left to the readers' interpretation, saying "results are shown in Table".

Reply: According to the author’s guidelines for correctly writing an original article, the section “Results” answers to the question “What did the study find? Was the tested hypothesis true?”, while the section “Discussion” to “What might the answer imply and why does it matter? What are the perspectives for future research?”.  For these reasons, we have reported the results as data with short descriptions and we have discussed the results in the section “discussion”. All the results are commented in detail in the discussion section, as a consequence the reader is therefore not led to interpret the data alone, but to critically analyse it according to our discussion.

I think that these aspects can be useful to increase the quality of presentation and scientific soundness.

Finally, in terms of references, some lacks seem to be present. I have suggested some exemplifying works which could be useful to broaden the views and provide further rigour to the treatise.

Reply: we have increased the number of references, limiting the choice to the most relevant ones considering the large number of articles published on the most important search engines (see “References”.

All the changes made are track in red in the revised manuscript in the attachment.

Reviewer 3 Report

The authors should do the following comments which will make the paper more suitable:

1- They should add a literature review section.

2- Extend the introduction section.

3- Rewrite the abstract and make it shorter.

4- Write more about the protocol, some steps are ambiguous.

5- Support Statistical analysis with figures. 

6- Add a section about lung ultrasonography (LUS), and show some examples and differentiation between the healthy and COVID-19.

7- Explain the used logistic regression algorithm, and add a subsection for it.

8- Plot figures for the logistic regression algorithm output. 

Author Response

The authors should do the following comments which will make the paper more suitable:

1- They should add a literature review section.

We have increased the number of references adding review articles on COVID-19 pandemic and LUS (see introduction in the attached file and the following references:

References

[8] Murgante B, et al.

 Why Italy First? Health, Geographical and Planning Aspects of the COVID-19 Outbreak. 

Sustainability, 12 (12) (2020), p.5064, 10.3390/su12125064.

[9] Comelli I, et al.

Impact of the COVID-19 epidemic on census, organization and activity of a large urban Emergency Department. 

Acta Biomed, 91(2) (2020), pp. 45-49, 10.23750/abm.v91i2.9565.

[10] Giostra F, et al.

Impact of COVID-19 pandemic and lockdown on emergency room access in Northern and Central Italy.

 Emergency Care Journal, 17 (2) (2020), 10.4081/ecj.2021.9705.

[11] Turcato G, et al.

 The COVID-19 epidemic and reorganisation of triage, an observational study. 

Intern Emerg Med, 15 (8) (2020), pp. 1517-1524,10.1007/s11739-020-02465-2.

[12] Carenzo L, et al.

Hospital surge capacity in a tertiary emergency referral centre during the COVID-19 outbreak in Italy [published correction appears in Anaesthesia. 2020 Nov;75(11):1540]. 

Anaesthesia, 75 (7) (2020), pp.928-934, 10.1111/anae.15072.

[13] Coen D, et al.

Changing Emergency Department and hospital organization in response to a changing epidemic. 

Emergency Care Journal, 16 (1) (2020), 10.4081/ecj.2020.8969.

[14] Barbieri G, et al.

COVID-19 pandemic management at the Emergency Department: the changing scenario at the University Hospital of Pisa. 

Emergency Care Journal, 16 (2) (2020), 10.4081/ecj.2020.9146

[15] Cerqua A, et al.

Local inequalities of the COVID-19 crisis.

 Regional science and urban economics, vol. 92 (2022): 103752, 10.1016/j.regsciurbeco.2021.103752.

[16] Dettori M, et al.

Air pollutants and risk of death due to COVID-19 in Italy.

 Environmental research, vol. 192 (2021): 110459, 10.1016/j.envres.2020.110459

[17] Giwa A L, et al.

Novel 2019 coronavirus SARS-CoV-2 (COVID-19): an overview for emergency clinicians.

Pediatric emergency medicine practice, vol. 17 (5) (2020), pp. 1-24

[18] Jeffery MM, et al.

Trends in Emergency Department Visits and Hospital Admissions in Health Care Systems in 5 States in the First Months of the COVID-19 Pandemic in the US. 

JAMA Intern Med, 180 (10) (2020), pp. 1328-1333, 10.1001/jamainternmed.2020.3288.

[19] Roberge D, et al.

The continuing saga of emergency room overcrowding: are we aiming at the right target?

Healthcare policy = Politiques de sante vol. 5,3 (2010), pp. 27-39.

[20] Pesaresi C, et al.

Emergency Department Overcrowding: A Retrospective Spatial Analysis and the Geocoding of Accesses. A Pilot Study in Rome. 

ISPRS International Journal of Geo-Information, 9 (10) (2020), p.579, 0.3390/ijgi9100579.

[21] Yarmohammadian MH, et al.

Overcrowding in emergency departments: A review of strategies to decrease future challenges.

 J Res Med Sci, 2017 (22) :23, 10.4103/1735-1995.200277.

[25] Caroselli C, et al.

Early Lung Ultrasound Findings in Patients With COVID-19 Pneumonia: A Retrospective Multicenter Study of 479 Patients.

Journal of ultrasound in medicine: official journal of the American Institute of Ultrasound in Medicine, 10.1002/jum.15944. 18 Jan. 2022, 10.1002/jum.15944.

[26] Abhilash Kundavaram P P, et al.

Acute management of COVID-19 in the emergency department: An evidence-based review.

Journal of family medicine and primary care, vol. 11 (2) (2022): 424-433, 10.4103/jfmpc.jfmpc_1309_21.

[27] Jackson K, et al.

Lung ultrasound in the COVID-19 pandemic.

Postgraduate medical journal, vol. 97 (1143) (2021), pp. 34-39, 10.1136/postgradmedj-2020-138137.

[28] Zhang Y, et al.

Lung Ultrasound Findings in Patients With Coronavirus Disease (COVID-19).

AJR. American journal of roentgenology, vol. 216 (1) (2021), pp. 80-84, 10.2214/AJR.20.23513.

 [29] Sorlini C, et al.

The role of lung ultrasound as a frontline diagnostic tool in the era of COVID-19 outbreak.

 Internal and emergency medicine, vol. 16 (3) (2021), pp. 749-756, 10.1007/s11739-020-02524-8.

 [30] Boccatonda A, et al.

One year of SARS-CoV-2 and lung ultrasound: what has been learned and future perspectives. 

J Ultrasound, 24 (2) (2022) :115-123, 10.1007/s40477-021-00575-x.

[34] Hosmer Jr, D. W., Lemeshow, S., & Sturdivant, R. X.

Applied logistic regression, Vol. 398 (2013). John Wiley & Sons.

2- Extend the introduction section.

Reply: We have done this change adding more considerations on the COVID-19 pandemic and its consequences in the organization of the EDs worldwide (see Introduction and References”). As requested by another reviewer, we described the city of Piacenza from a geolocation point of view and reported the data on accesses to our emergency room in the first month of the pandemic.

3- Rewrite the abstract and make it shorter.

Reply: done (word count: 210).

Background: COVID-19 pandemic has forced the emergency departments (ED) to reorganize their working methods and find real solutions in such a complex situation. We developed a score based on the lung ultrasonography (LUS) and the pre-triage interview to correctly manage the patients in the ED using a three-path organisation (COVID-19, not-COVID-19 and intermediate). Methods: We retrospectively analysed 292 patients admitted to our ED from April 10th to April 15th, 2020. We identified a set of predictors of SARS-CoV-2 infection selected from the pre-triage interview items and the LUS findings, and we compared the two different pathways organizations. Results: Nasopharyngeal swab positivity, close contact with a COVID-19 patient, fever with respiratory symptoms, respiratory failure, anosmia or dysgeusia, diffuse alveolar interstitial syndrome, absence of B lines and presence of pleural effusion are the most informative elements for classifying the patient. The most significant difference between the two pathways is the percentage of not-COVID-19 patients assigned to the COVID-19 area: 10.6% in the three-path organisation and 18.9% in the two-path organisation (p =0,037). Conclusion: Our study demonstrates that a simple score can be useful to correctly manage the patients admitted to the ED, and the intermediate area can limit the spread of SARS-CoV-2 in the ED and as a consequence in the hospital.

4- Write more about the protocol, some steps are ambiguous.

Reply: we ask the reviewer to kindly indicate ambiguous and unclear points in the protocol so that they can be correctly identified and easily clarified.

5- Support Statistical analysis with figures. 

Reply: We added a figure to display the algorithm output (see point 8).

6- Add a section about lung ultrasonography (LUS), and show some examples and differentiation between the healthy and COVID-19.

Reply: LUS in a common technique used in the emergency department and extremely important fort the management of respiratory failure as reported in literature, not only for COVID-19 pneumoniae. We believe that it is rather pleonastic to describe normal lung ultrasound as this method has been widely used in the emergency setting for several years and is a required skill for all emergency physicians. For our protocol we have used the model reported by Volpicelli G, et al. Intensive Care Medicine, 38 (4) (2012), pp. 577-91, 10.1007/s00134-012-2513-4 (see materials and method for the reference), that clearly understand how to perform a LUS (figure 1) and all the possible findings (figure 2-5) including how to calculate the LUS score (0-3). In addition, in the introduction we have reported a letter to the Editor by Poggiali et al. and an original article by Dacrema et al. (both the authors belong to the Emergency Department of Piacenza) who clearly explain the role of LUS and the different LUS pattern in COVID-19 pneumoniae.

7- Explain the used logistic regression algorithm, and add a subsection for it.

Reply: We added the following subsection in Statistical Analysis: “The logistic regression algorithm is used to express a response variable as a function of a set of possible predictors and an idiosyncratic residuals when the response variable is binary. It is similar to linear regression where the mean response variable, say , is expressed a linear function of possible predictors , i.e.

Nonetheless it is different as when y is binary (0/1) its expectation is a probability ( and as such a number in the [0,1] interval. A linear combination such as could generate predicted values outside this range.

For this reason a transformation of  is used instead, namely we have  where  stays for the logarithm of the probability odd of  i.e. .

The  parameters are unknown and estimated from the data using an algorithm known as iterative least square to maximize the data likelihood.

Each  describes how of changes when  increases by 1 unit, all the remaining predictors kept constant; describes how the oddchanges under the same conditions.

(Hosmer Jr, D. W., Lemeshow, S., & Sturdivant, R. X. (2013). Applied logistic regression (Vol. 398). John Wiley & Sons).

8- Plot figures for the logistic regression algorithm output.

Reply: We added the following figure (see the file in the attachment)

Figure 4. Probability of being a COVID-19 patient predicted by the logistic regression model is plotted against the actual patient status.

Round 2

Reviewer 2 Report

The paper has acquired added value if compared with the previous proof. I think that some other efforts could have done regarding critical contextualization also with the support of suggested works and with some other specific figures (among other things the added figure 1 is very simply) and related comments.

Author Response

We thank the reviewer 2 for the comments.

We have tried to improve the quality of the figure 1 and to better describe the Italian situation during COVID-19 pandemic using some of the references suggested by the reviewer (the actual number of the references is 44), even if the main aim of our manuscript is not to describe a well-known and dramatic situation as COVID-19 Italian epidemic was, as reported by several other Italian authors in the current literature, including our emergency team (references: [5] Poggiali E, et al.  COVID-19 pandemic, Piacenza calling. The survival strategy of an Italian Emergency Department.Acta Biomed, 91 (3) (2020), p. 2020045, 10.23750/abm.v91i3.9908; [6] Maniscalco P, et al.  The deep impact of novel CoVID-19 infection in an Orthopedics and Traumatology Department: the experience of the Piacenza Hospital. Acta Biomed, 91 (2) (2020), pp. 97-105. 10.23750/abm.v91i2.9635; [7] Poggiali E, et al.  Triage decision-making at the time of COVID-19 infection: the Piacenza strategy Intern Emerg Med, 15 (5) (2020), pp. 879-882. 10.1007/s11739-020-02350-y). We want to describe a method to correctly manage patients in the emergency department to avoid the possible spread of the virus, even among negative patients, using a simple and easy method based on the pre-triage interview and LUS.

We have also described better in the introduction the role of point-of-care lung ultrasound (LUS) adding some important references in order to improve the quality of this paper and help the reader to better understand the importance of LUS in the diagnosis and management of COVID-19 pneumoniae, as reported in literature. We have added the figure 3 to better explain the lung damage documented by the LUS in COVID-19 patients, compared to HRCT imaging.

Counting all the figures and tables, there are respectively 5 figures and 9 tables, that we consider a significant number for an original article. Anyway, if the reviewer 2 wants other figures, we need to know exactly which kind of figures are requested.

The modifications/incorporations can also be traced in track changes in main document.

Reviewer 3 Report

Dear Authors,

Thank you very much for your constructive replies and modifications to the paper. I think now the paper is ready for publication.

Author Response

Thanks for your comments.